# Development and Characterization of Chitosan Microparticles-in-Films for Buccal Delivery of Bioactive Peptides

**DOI:** 10.3390/ph12010032

**Published:** 2019-02-20

**Authors:** Patrícia Batista, Pedro Castro, Ana Raquel Madureira, Bruno Sarmento, Manuela Pintado

**Affiliations:** 1Escola Superior de Biotecnologia, Centro de Biotecnologia e Química Fina, Rua Arquiteto Lobão Vital, 172, 4200-374 Porto, Portugal; pbatista@porto.ucp.pt (P.B.); pedro.joao.castro@gmail.com (P.C.); rmadureira@porto.ucp.pt (A.R.M.); 2CESPU, Instituto de Investigação e Formação Avançada em Ciências e Tecnologias da Saúde, Rua Central de Gandra 1317, 4585-116 Gandra-PRD, Portugal; bruno.sarmento@ineb.up.pt; 3i3S—Instituto de Investigação e Inovação em Saúde, Universidade do Porto, Rua Alfredo Allen 208, 4200-393 Porto, Portugal; 4INEB—Instituto Nacional de Engenharia Biomédica, Universidade do Porto, Rua Alfredo Allen 208, 4200-393 Porto, Portugal

**Keywords:** bioactive peptides, buccal delivery, chitosan, microparticles, oral films

## Abstract

Nowadays, bioactive peptides are used for therapeutic applications and the selection of a carrier to deliver them is very important to increase the efficiency, absorption, release, bioavailability and consumer acceptance. The aim of this study was to develop and characterize chitosan-based films loaded with chitosan microparticles containing a bioactive peptide (sequence: KGYGGVSLPEW) with antihypertensive properties. Films were prepared by the solvent casting method, while the microparticles were prepared by ionic gelation. The final optimized chitosan microparticles exhibited a mean diameter of 2.5 µm, a polydispersity index of 0.46, a zeta potential of +61 mV and a peptide association efficiency of 76%. Chitosan films were optimized achieving the final formulation of 0.79% (*w*/*v*) of chitosan, 6.74% (*w*/*v*) of sorbitol and 0.82% (*w*/*v*) of citric acid. These thin (±0.100 mm) and transparent films demonstrated good performance in terms of mechanical and biological properties. The oral films developed were flexible, elastic, easy to handle and exhibited rapid disintegration (30 s) and an erosion behavior of 20% when they came into contact with saliva solution. The cell viability (75–99%) was proved by methylthiazolydiphenyl-tetrazolium bromide (MTT) assay with TR146 cells. The chitosan mucoadhesive films loaded with peptide–chitosan microparticles resulted in an innovative approach to perform administration across the buccal mucosa, because these films present a larger surface area, leading to the rapid disintegration and release of the antihypertensive peptide under controlled conditions in the buccal cavity, thus promoting bioavailability.

## 1. Introduction

In the last decade, protein and peptide delivery has become an important area of research for therapeutic applications [1,2,3,4]. Bioactive peptides are defined as specific protein fragments that have a positive impact on body functions or conditions, presenting many beneficial health effects (e.g., antimicrobial, antioxidant, antithrombotic and antihypertensive properties) [5,6]. The activity of bioactive peptides is based on their inherent amino acid composition and sequence. The peptide sequence KGYGGVSLPEW was identified in a whey protein hydrolysate and was recognized as a bioactive peptide with antihypertensive properties [7]. This peptide can be administered by parenteral route to avoid biological barriers that can hinder permeability. Nonetheless, the oral delivery of peptides and proteins remains an easier, more attractive and convenient alternative [8,9]. The buccal mucosa represents an important non-invasive alternative route for protein and peptide delivery, due to its ease of administration, evasion from the first pass hepatic metabolism, high relative permeability of many therapeutic agents and rich vascularization [10,11].

However, a number of factors limit the absorption of peptides, due to their relatively large molecular size, physical/chemical barriers, involuntary swallowing of dosage forms and continuous dilution of dissolved molecules by saliva [2]. Indeed, the absolute buccal bioavailability levels of most peptides and proteins are less than 1% [6,12,13]. So, numerous approaches have been attempted with the aim of improving the permeability of peptides. The development of delivery systems such as micro/nanoparticles (MPs/NPs) and their use as carriers coupled to other systems represents a valid approach to overcome these drawbacks. Such systems have attracted growing scientific and commercial attention as bioactive protein/peptide carriers during the last few years [6,14]. The encapsulation of bioactive peptides presents many advantages, such as improved efficiency and absorption, enhanced protection from enzymatic and pH degradation, controlled release of loaded peptide, increased bioavailability and enhanced patient compliance [10,14,15]. The development of nano/micro carriers and the use of the buccal route for mucosal (local) and transmucosal (systemic) delivery of therapeutic macromolecules is an interesting and promising combination. The association of mucoadhesive oral films with MPs/NPs represents a promising strategy to overcome these obstacles [10].

In the development of these delivery systems, polymers are often used as a matrix for peptide loading. Thus, chitosan (CH) was chosen as the polymer to be used, because previous studies have shown it to possess diverse biological activities, including biocompatible, biodegradable, non-toxic, antihypertensive, anti-inflammatory and antimicrobial properties, and it has shown great potential in applications of drug delivery [14,16,17,18,19]. Chitosan is a mucoadhesive polymer due to its ability to form ionic, pH-dependent interactions with mucin and its ability to enhance the penetration of large molecules across the mucosal surface. Therefore, it is a good candidate for buccal delivery [20,21]. So, chitosan has attracted attention as a potential food preservative of natural origin and was approved by the United States Food and Drug Administration (USFDA) as a Generally Recognized as Safe (GRAS) food additive and as a delivery system to the human body, more specifically as an oral delivery system [22].

This paper reports the development and characterization of chitosan microparticles loaded with an antihypertensive peptide with subsequent incorporation into chitosan films, aiming to achieve the administration of peptides across the buccal mucosa. This delivery system (film-MPs) is interesting because the mucoadhesive films administered to the mucosal surface could represent a delivery device with multifunctionalities such as mucoadhesion, control release and drug protection by avoiding or reducing passage through the gastrointestinal tract, and a conveyor system of MPs with bioactive peptides. The MPs exert their function by protecting the bioactive peptides from degradation and acting as a controlled release system because chitosan mucoadhesive properties are able to promote enhanced bioactive molecules delivery.

This study intends to have an impact on the nutraceutical and pharmaceutical industry. The development of an oral film incorporating microparticles enhances the advantages of oral films in terms of administration (reported in many studies) with the controlled delivery. The fact that the transported molecule is a peptide resulting from whey protein hydrolysate enhances its nutritional value and associated therapeutic potential. Therefore, this new delivery system may prove to be an enhancer of controlled release delivery of bioactive molecules with potential therapeutic effect, although clinical studies are needed.

## 2. Materials and Methods

### 2.1. Materials and Cell Line

KGYGGVSLPEW peptide was purchased from GenScript (Piscataway, USA). Low-molecular-weight chitosan (50,000–190,000 Da, 75–85% of deacetylation), ethyl acetate, pentasodium tripolyphosphate (TPP), α-amylase, pepsin, bovine bile salts, pancreatin, trifluoroacetic acid (TFA) and D-sorbitol (assay purity ≥ 98%) and phosphate-buffered saline tablets (PBS) were purchased from Sigma-Aldrich (Steinheim, Germany). Citric acid monohydrate, potassium phosphate monobasic anhydrous and sodium phosphate dibasic were obtained from Merck (Darmstadt, Germany). Sodium chloride was purchased from Panreac (Barcelona, Spain). Glacial acetic acid, sodium hydroxide and all other chemicals were of analytical grade, purchased from Sinopharm Chemical Reagent Co., Ltd., Shanghai, China. Methanol and acetonitrile (HPLC gradient grade) were purchased from Fisher (Loughborough, UK). Ultrapure water was used to prepare all formulations. 

TR146 cell line was purchased from Sigma-Aldrich (Stenheim, Germany). Fetal bovine serum (FBS), HAMS-F12 culture medium and Pen-Strep (10,000 U Penicillin, 10,000 U Streptomycin) were purchased from Lonza^®^ (Verviers, Belgium). TrypLE^TM^ express was purchased from Gibco^®^ (Taastrup, Denmark). Thiazolyl Blue Tetrazolium Bromide (MTT) was purchased from VWR (Solon, USA). Dimethyl sulphoxide (DMSO) 99.7% was purchased from Fisher Bioreagents^TM^ (Pennsylvania, USA). Lastly, 96-well plates were purchased from Thermo Scientific (Hvidovre, Denmark). 

### 2.2. Formulation of Chitosan Microparticles Loaded with the Antihypertensive Peptide

Chitosan microparticles (CH MPs) were prepared by the ionic gelation method of chitosan with TPP [23]. Chitosan solutions were prepared by dissolving 40 mg of chitosan (0.4–0.1 M) in 2 mL of a 1% (*v/v*) glacial acetic acid solution [24]. Afterwards, the peptide was dissolved in the chitosan solution and 1.5 mg of TPP (cross-linker) was added and left under magnetic stirring at 1000 rpm for 90 min at room temperature. Microparticles were formed spontaneously upon the incorporation of TPP into the CH solution.

#### Preliminary Optimization and Factorial Design

Experimental design for chitosan microparticles was performed using SAS JMP^®^ 9 software [25]. Factorial design allowed all the factors to be varied simultaneously, enabling the evaluation of the effects of each variable at each level and showing the interrelationship among them. The number of experiments required for these studies was dependent on the number of independent variables selected. Each design was performed considering five dependent variables (size, zeta potential, polydispersity index (PDI), association efficiency and loading degree) as well as two independent variables (polymer and TPP concentration). Every response test was performed in triplicate. When responses were determined, independent variables that influenced the behavior of evaluated dependent variables were selected for the elaboration of the predictive statistic model, according to RSquare and RSquare adjusted values. Finally, optimal formulations for each polymer were obtained by maximizing desirability values.

### 2.3. Characterization of Chitosan Microparticles

After production, MPs were characterized for their mean particle size and PDI by dynamic light scattering. Zeta potential was determined by phase analysis light scattering. All measurements were performed using a Malvern Zetasizer Nano ZS instrument (Malvern Instruments Ltd., Malvern, UK). For these measurements, samples were diluted in saline solution.

### 2.4. Association Efficiency and Loading Degree

The CH MPs association efficiency was determined upon the separation of MPs from the aqueous preparation medium containing the non-associated protein by centrifugation (15,000× *g*, 45 min, 15 °C). The amount of free peptide was determined in the supernatant by a HPLC-UV (Waters Alliance^®^ instrument (Milford, MA, USA)) method. In this method, a Kromasil^®^ C18 column (AkzoNobel, Bohus, Sweden) was used and the UV detector wavelength was set to 280 nm. The mobile phases consisted of acetonitrile and 0.1% TFA, and water and 0.1% TFA. The ratio was initially set at the ratio of 80:20 (acetonitrile: 0.1% TFA, *v*/*v*), which linearly changed to a 40:60 (*v*/*v*) gradient over 10 min. The flow rate was 0.8 mL/min and the injected volume of the sample was 20 μL. The UV detector wavelength was set at 280 nm. The total area under the peak was used to quantify the KGYGGVSLPEW peptide sequence. Each sample was assayed in triplicate (*n* = 3). The CH MPs peptide association efficiency (AE) and loading degree (LD) were calculated as follows (Equations (1) and (2)):(1)$$AE\;\left(\%\right)=\frac{total\;peptide\;amount-free\;peptide\;amount}{total\;peptide\;amount}\times 100,$$
(2)$$LD\;\left(\%\right)=\frac{total\;peptide\;amount-free\;peptide\;amount}{peptide\;loaded\;CH\;MPs\;dry\;weight}\times 100,$$

### 2.5. Preparation of Chitosan Oral Films

Chitosan films were prepared by the solvent casting method with some modifications [26]. The composites (78.6 mg chitosan (0.15–0.04 M), 82.5 mg of citric acid (0.039 M) and 674 mg of sorbitol (0.37 M)) were added to 10 mL of deionized water. The mixture was covered and stirred (magnetic stirring, 300 rpm, at room temperature for 120 min) until chitosan was totally dissolved. Subsequently, the solution (10 mL) was dispensed into Petri dishes (90 × 15 mm) and dried for 48 h in an incubator set to 30 °C. After drying, films were then cut into squares (2 × 2 cm).

#### Chitosan Films Experimental Design Testing

The experimental design employed SAS JMP^®^ 9 software (JMP Statistical Discovery^TM^, Marlow, UK), using a similar procedure to that described in microparticles section. Each design was performed considering five dependent variables: elongation at break, tensile strength, Young’s modulus, water uptake and erosion. For each polymer tested, three independent variables were considered (polymer concentration (chitosan 0.5, 1, 1.5 (%, *w*/*v*)), plasticizers (sorbitol) concentration (32.5, 56.3, 75 mg/mL) and citric acid concentration (7.5, 10, 12.4 mg/mL)). Citric acid was used to induce the production of saliva in order to promote the disintegration of oral films the oral cavity [15]. Sorbitol concentration was stipulated according to polymer dry weight.

Each sample was tested in triplicate (*n* = 3). When the results were obtained, a screening design was executed, and independent variables that influenced the behavior of the evaluated dependent variables were selected for the elaboration of the predictive statistic model (with RSquare and RSquare adjusted values). Thus, optimal formulations were obtained by setting desirability values to each response type to obtain maximum desirability.

### 2.6. Chitosan Films Characterization

#### 2.6.1. Film Appearance

The appearance of films was evaluated by visual observation using parameters such as the transparency and semi-transparency nature of the strip [26]. 

#### 2.6.2. Film Weight and Thickness

Films strips were weighed on a calibrated analytical scale and the thickness was measured using a calibrated Vernier gauge caliper micrometer [15].

#### 2.6.3. Determination of the Mechanical Properties

The main mechanical properties such as tensile strength (MPa), strain at tensile strength (%), Young’s modulus and strain energy (MPa) (Equations (3) and (4), respectively) were evaluated. For that purpose, the developed films were cut in squares (2 × 2 cm) and these properties were measured using a texturometer (TA.XT plus Texture Analyser, Stable Micro Sydtems, Cardiff, UK) [15]. All measurements were performed in three films for each formulation.
(3)$$Young^{\prime }s\;modulus\;\left(MPa\right)=\frac{Force\;at\;corresponding\;strain}{Cross-sectional\;area\;of\;the\;film\;\times \;Corresponding\;strain}\times 100,$$
(4)$$Strain\;Energy\;\left(MPA\right)=\frac{1}{2}\times \frac{volume}{Young^{\prime }smodulus}\times Stress^{2},$$

#### 2.6.4. Swelling and Erosion Studies

Plain films were characterized for their swelling properties and erosion features by calculating the percentage of hydration and matrix erosion of the films. Films (2 × 2 cm) were cut and weighed (*W*_0_). Subsequently, films were immersed in the artificial salivary solution (pH 6.8) for a consecutive series of 30 s each, over 1 min. At these time intervals, the films were wiped off using filter paper and weighed (*W*_1_). The swelling of the films was determined using the following relation (Equation (5)): (5)$$Swelling\;\left(\%\right)=\frac{W_{1}-W_{0}}{W_{0}}\times 100,$$
where *W*_1_ is the weight of swollen film after time t and *W*_0_ is the weight of the film at time zero.

After complete hydration, films were dried at 37 °C for 24 h. The dried films were taken and their weight was registered (*W*_2_). Erosion was calculated using the following relation (Equation (6)):(6)$$Erosion\;\left(\%\right)=\frac{W_{1}-W_{2}}{W_{1}}\times 100,$$
where *W*_1_ is the weight of swollen film after time t and *W*_2_ is the weight of dry film after erosion.

### 2.7. Chitosan Films with Chitosan Microparticles

Chitosan MPs (0.4–0.1 M) were incorporated into CH film solutions (15.7 mg of chitosan (0.157–0.041 M), 16.5 mg of citric acid (0.039 M) and 134.8 mg of sorbitol (0.37 M) in 2 mL of deionized water for 30 min in order to uniformly disperse MPs. The CH films with CH MPs were prepared by the solvent casting method. 

Chitosan films with CH MPs were poured on Petri dishes and placed to dry for 48 h in an incubator at 37 °C. After drying, films were cut into squares (2 × 2 cm). Each formulated film was prepared in triplicate.

### 2.8. Cell Culture

Cell culture systems are important for the examination the biological properties, such as the bioavailability or toxicity of bioactive molecules. The TR146 cell culture model was selected as an in vitro model of the human buccal epithelium. The TR146 cell line originated from a human buccal carcinoma. After culturing, the TR146 cell line forms a stratified epithelium similar to the buccal epithelium [27,28].

The TR146 cells were grown in HAMS F-12 Medium with supplements of 10% (*v*/*v*) fetal bovine serum (FBS) and 1% (*v*/*v*) antibiotic/antimitotic mixture (final concentration of 100 U/mL penicillin and 100 U/mL streptomycin). Culture conditions were maintained at 37 °C, 5% CO_2_ and 95% relative humidity. Sub-cultivation was performed at approximately 80% confluence with 0.25% trypsin-EDTA to detach the cells from the flasks. Cells were then seeded at a density of 1 × 10^6^ cells per 75 cm^2^ flask. The culture medium was replaced every other day. Cells were maintained in an incubator (BB 16 gas incubator, Heraeus Instruments GmbH) at 37 °C, 5% CO_2_ and 95% relative humidity. 

#### Cell Viability Studies

The cell viability of TR146 cell line, after 24 h treatment with CH MPs, CH films, with or without peptide (with concentration 5 μg/mL), was measured using the methylthiazolydiphenyl-tetrazolium bromide conversation (MTT) assay [23].

Cells were seeded in 96-well plates at 2 × 10^5^/well in 300 μL culture medium and incubated for 24 h at 37 °C in a 5% CO_2_ environment. The medium was then changed and the cells were treated with test samples (peptide, CH MPs, CH MPs with peptide; CH films; CH films with peptide; CH films with MPs with peptide or free) for 24 h. Each treatment was tested in six individual wells. After 24 h, the supernatant was removed and 200 μL of MTT solution (5 mg/mL in the cellular culture medium) was added to each well of the 96-well plates. They were then incubated for 4 h at 37 °C to allow the formation of formazan crystal. The medium was then removed, and the blue formazan was eluted from cells using 150 μL of DMSO. The negative control used was also DMSO. The plates were shaken on an orbital shaker to solubilize the crystals of formazan. The dark blue crystals were aspirated to another new 96-well microplate and the optical density (OD) was measured directly in the microplate reader at 570 and 690 nm for background reduction. All samples were tested for *n* = 5 experiments with comparable results.

The cell viability of the tested delivery systems was calculated from the average OD values (Equation (7)).
(7)$$Cell\;viability\;\left(\%\right)=\frac{OD\;value\;of\;specimen\;suspension}{OD\;value\;of\;negative\;control\;suspension}\times 100,$$

### 2.9. Statistical Analysis

Statistical analysis was performed using SPSS^®^ for Windows version 22 (IBM SPSS, Chicago, IL). The average percentage of peptide released from CH films was calculated for each time point, along with respective standard deviation values.

The *t*-test was used to verify the existence of statistically significant differences between predictive models and experimental results. Chitosan MPs experimental data were obtained from three samples and the mean values were compared with the values predicted in the model.

Prediction formulas that describe the statistically significant influence of independent variables on dependent variables were obtained using SAS JMP^®^ software. From the analysis of RSquared and adjusted RSquared, the best models were chosen and prediction formulas were obtained. From the predictive models, predictive profilers were obtained and optimal formulations were determined for each formulation of the CH oral film.

## 3. Results and Discussion

### 3.1. Preparation and Characterization of Chitosan Microparticles

Chitosan MPs were prepared by the ionic gelation method by auto-aggregation between a positively charged amino group of chitosan and the negatively charged phosphate groups of TPP (cross-linking agent) [19,29,30]. Chitosan MPs cross-linked with TPP have been employed in many studies for drug delivery systems because TPP is used to improve the mechanical properties and stability of CH MPs [8,29]. The parameters, type of cross-linking agent and polymer were optimized in preliminary studies. The CH MPs optimized formulation was set as 40 mg chitosan (polymer) and 1.5 mg TPP (cross-linker). Figure 1 outlines the factorial design and values of the formulation parameters. 

In order to achieve theoretical optimization to validate the results, the formulations of CH MPs were further assessed for mean size, polydispersity index, zeta potential and association efficiency. The obtained results were individually compared with the theoretical (predicted) values by Student *t*-tests. No statistically differences (*P* > 0.05) were found between predicted and experimental values.

After the optimization of the CH MPs, peptide-loaded CH MPs were prepared and parameters such as particle size, zeta potential, PDI, association efficiency and loading capacity were analyzed because those properties are important for therapeutic properties. 

The mean size of CH MPs is dependent on both chitosan molecular weight and concentration and on TPP concentrations. Particle size can influence the biopharmaceutical properties of microparticles, their biodistribution and the particle content uptake [8,31]. Particle size was the leading assessed property during formulation optimization studies, oriented towards obtaining microparticles with a mean diameter of about 2.5 µm with a reproducible size distribution. As shown in Table 1, unloaded CH MPs presented a size of 2.544 ± 0.97 μm and peptide-loaded CH MPs had a size 2.582 ± 0.87 μm. The determined size of chitosan microparticles was in agreement with results reported in the literature [32].

Peptide loading by the encapsulation method did not induced an increase in particle size when compared with empty CH MPs (Table 1). So, the incorporation of the peptide into CH MPs did not have a significant effect on particle size. 

The particle size and surface charge of MPs/NPs regulate the biodistribution and pharmacokinetic properties of the MPs/NPs in the body. Therefore, the zeta potential is another important parameter and useful indicator of the electronic charge, which can be used to predict and control the stability of colloidal suspensions or emulsions [8,31]. The greater the zeta potential, the more likely the suspension is to be stable because the charged particles repel one another and thus overcome the natural tendency to aggregate. Microparticles with a zeta potential above ± 30 mV have been shown to be stable in suspension, as the surface charge prevents the aggregation of the particles [33]. According to the results obtained (Table 1), all the batches prepared showed a zeta potential more than + 30 mV, confirming that microspheres exhibited good stability and no aggregation in the suspension. The positive value of the zeta potential might be due to the positive charge of chitosan and the high positive zeta potential indicated that the electrostatic repulsion between particles prevented aggregation and increased their stability. The positive value of the zeta potential is important for buccal drug delivery since it can facilitate adhesion to the mucosal epithelial surface, thus prolonging the peptide release and enhancing the peptide bioavailability. The results showed that the addition of peptide has no significant effect on the microparticles zeta potential.

The PDI values of CH MPs and peptide-loaded CH MPs were around 0.5 (the index is a measure of dispersion homogeneity; values closer to zero indicate a homogeneous dispersion), indicating uniformity of particle size and monodispersity distribution, with low variability and no aggregation, as reported in the literature [30,34]. If a scale from 0 to 1 is considered, a PDI lower than 0.1 might be associated with a high homogeneity in the particle population, whereas high PDI values suggest a broad size distribution or even several populations. The calculation of PDI takes into account the mean particle size, the refractive index of the solvent, the measurement angle and the variance of the distribution. So, the PDI affects the mechanical strength of the polymer and its ability to be formulated as a delivery device, and these properties may control the polymer biodegradation rate [35].

Association efficiency and loading capacity are other characteristics that should be calculated for controlled delivery systems [8]. The association of bioactive peptides with the delivery systems components conditions the delivery system success, because it can protect biomolecules against metabolic degradation and improve protein absorption into the intestinal epithelium with better bioavailability. CH MPs were successfully prepared via the ionic gelation method and ensured encapsulation of the peptide. Although the association efficiency of hydrophilic molecules is usually low, in this study we obtained high encapsulation efficiency values, similar to other studies. Table 1 shows the association efficiency and loading capacity of peptide-loaded CH MPs. The CH MPs with peptide showed an encapsulation efficiency of 76%, achieving a particle loading degree of 0.46% (*n* = 3). The antihypertensive peptide was successfully entrapped into the CH MPs with a high association efficiency, indicating the good potential of CH MPs as a delivery system. The AE was optimized by varying some parameters, including the amount of chitosan and TPP concentrations. The AE and size are important indexes for evaluating the quality of delivery systems. The high AE% can improve the utilization of the peptide and a smaller size could enhance the absorption of buccal cells [36,37]. Indeed, other authors [33,36,38] have already proven that CH MPs are natural materials with excellent physicochemical properties, good carriers for encapsulating proteins, which can achieve high protein loading efficiency and protect them from degradation.

### 3.2. Chitosan Films Characterization

Various methods have been described in the literature as appropriate to prepare CH films for delivery systems [26]. The solvent casting method was selected because it is the method most commonly reported in the literature due to its inherent simplicity and robustness. It is a feasible and cost-effective technique which ensures greater commercial viability.

Firstly, a CH film experimental design was performed in order to obtain optimized formulations and understand how excipients influence the mechanical characteristics of the films. Figure 2 shows the prediction profilers used in the optimization of the formulations of CH films.

By setting the desirability of dependent variables to maximum, it was possible to obtain the best possible formulations. So, the optimized formulation of the CH films was set as 0.79% (*w*/*v*) of chitosan (polymer), 6.74% (*w*/*v*) of sorbitol (plasticizer), 0.82% (*w*/*v*) of citric acid (salivary stimulator) and an ideal thickness of 136 μm.

Chitosan is commonly used for producing MPs/NPs thanks to their excellent properties (biocompatible, biodegradable, non-toxic, antihypertensive, anti-inflammatory, antimicrobial, mucoadhesive). However, the properties of these carriers can be further improved by the addition of plasticizers; for example, in this case sorbitol was used. The plasticizer molecules interposing between the polymer chains and interacting with their functional groups increased polymer chain mobility and flexibility and improved mechanical properties. Specifically, they reduced brittleness, improved flow, imparted flexibility and increased the toughness of films [17]. 

For the preparation of the films, citric acid was also added as a saliva stimulating agent [26]. The purpose of using citric acid is to increase the rate of production of saliva, which would aid in the faster disintegration and consequently the rapid dissolution of the film.

After optimized CH films composition, the films were produced at the same time as CH MPs. That way, the constituents of the films were added to the microparticle solution. Finally, the final solution was dried at room temperature and the optical and morphological properties were evaluated. These films were transparent, flexible and homogeneous, and their surfaces appeared to be smooth without pores and cracks [39,40]. The films were thin, with a thickness ranging between 0.085 and 0.117 mm, evaluated using a digital Vernier caliper [40]. The thickness, flexibility, elasticity and easy handling are important properties for oral films application and consumer acceptance [17]. So, we needed to evaluate mechanical properties: the elastic modulus, to evaluate the film’s rigidity; the tensile strength, to determine the brittleness of the film; the elongation at break, to know the flexibility and elasticity. These properties needed to be investigated as they condition the film’s integrity and its performance [38]. The Young’s modulus, tensile strength and elongation at break were measured and are shown in Table 2. 

The tensile strength is an important mechanical property to avoid damage (release of the carrier molecule) during post-production storage and transporting. Basically, this test is performed to measure of the maximum strength of a film to withstand applied tensile stress, and the percent elongation represents the ability of a film to stretch [41]. In optimized CH films, the tensile strength obtained was 0.767 ± 0.091 (MPa). This result is very low when compared with the tensile strength of the pure CH film in other studies, such as 8 MPa [41], 10.97 MPa [40] and 98 MPa [39]. The different results may be due to differences in chitosan type, plasticizer presence, film formation method or analytical methods used [39,40].

Another mechanical property is the Young’s modulus, which is an indicator of the stiffness (rigidity) of the film. It is reported to offer a sharp burst release of carrier molecules. The elongation at break is an indicator of its extensibility. The Young’s modulus for CH films incorporated with peptide-loaded CH MPs are higher than CH films (Table 2), because the Young’s modulus increased with the increase of filler content [42], but no statistically significant differences (*P* > 0.05) were found. The values obtained were low when compared with those of other oral films [43], but the composition and the evaluation methods were not the same. The low Young’s modulus obtained in our films indicates softer networks, lower water sorption and higher solubility.

The tensile strength and elongation at break values obtained for the peptide-loaded CH MPs indicate that the incorporation of the peptide into the films did not significantly alter the tensile strength compared with that of uncoated films, corroborating the findings of Aguilar and collaborators [44]. 

Films were also analyzed regarding disintegration capacity. Effectively, orodispersible films have a high delivery potential because when placed on the tongue, they are immediately hydrated by saliva, followed by disintegration and/or dissolution and the release of the bioactive peptide [45]. This CH film with peptide-loaded CH MPs, when contacted with saliva solution, showed a quickly swelling (217.05 ± 122.36%) and erosion (17.25 ± 12.21%) due to the disentanglement of the loosely bound chitosan molecules, which allowed a facile diffusion of the peptide-loaded CH MPs from the matrix (see Table 3) [46]. The swelling of the films first increased dramatically due to the porous structure and the hydrophilicity of the CH film, indicating a strong hydration of chitosan, which facilitates the rapid mucoadhesion to the absorptive epithelia [46]. Mucoadhesion occurs when the CH film comes in contact with buccal epithelial cells; a double layer of electrical charge forms at the interface to promote the adhesion [46]. The data obtained in this study confirm other reports in the literature; that is, the optimized CH film provides rapid disintegration (30 s) and the release of actives when the strip comes into contact with saliva in the mouth. These results agree with the range of values indicated by the Guidance for Industry [47,48]. 

### 3.3. In Vitro Cell Viability

In addition to the preparation and characterization, the in vitro evaluations of CH films with peptide-loaded CH MPs are important for understanding the behavior of these delivery systems in biological systems, as well as for elucidating the nature of interaction between the delivery system and tissues, i.e., the biocompatibility. Among the biocompatibility tests, cytotoxicity is preferred as it is simple, fast and has a high sensitivity. The cytotoxicity test is one of the most important methods for biological evaluation. In order to evaluate the cytotoxicity of developed formulations, the MTT assay was performed. The effect of CH films with peptide-loaded CH MPs on TR146 cells was tested in vitro.

Cytotoxicity ratios and viability were classified according to the following criteria for cytotoxicity: (a) if viability > 100%, the corresponding cytotoxicity type was class 0, indicating no toxicity; (b) if viability = 0%, the corresponding cytotoxicity type was class 5, indicating the highest toxicity; (c) 75–99%, 50–74%, 24–49% and 1–25% viability were categorized as classes 1, 2, 3 and 4, respectively [49].

Figure 3 outlines the results obtained from the MTT assay of TR146 cells after being exposed to developed formulations for a period of 24 h. The results indicat that all experimental conditions (CH MPs; peptide-loaded CH MPs; CH films; peptide CH films; CH films with CH MPs; CH films with peptide-loaded CH MPs) assured high cell viability. Hence, the cytotoxicity could be categorized as class 1 (Figure 3), which demonstrates that the CH films with peptide-loaded CH MPs have excellent cell biocompatibility. The results were obtained in conformity with other studies in which chitosan did not interfere with cell viability [20].

## 4. Conclusions

The present study presented an innovative approach focused on combining CH MPs with CH films as delivery systems for the potential administration of peptide/proteins across the buccal mucosa. CH MPs loaded with antihypertensive peptide were successfully prepared using the ionic gelation method and showed desirable pharmaceutical properties including small size and high AE%. Encapsulation of the peptide into CH MPs enhanced the peptide stability and the controlled release. 

Chitosan MPs with peptide were incorporated into CH films by solvent casting. The method used for film production is practical, simple, safe and reproducible. These mucoadhesive films have the ability to enhance the penetration of large molecules across the oral mucosal surface and have a protective effect of the MPs, which can elevate the peptide bioavailability.

Therefore, the developed system has ample potential for the delivery of drugs or/and bioactive molecules (bioactive peptides) by the oral route. It constitutes a good solution for the oral delivery of antihypertensive small peptides, and might be a potential strategy for hypertension control in the future.

## Figures and Tables

**Figure 1 pharmaceuticals-12-00032-f001:**
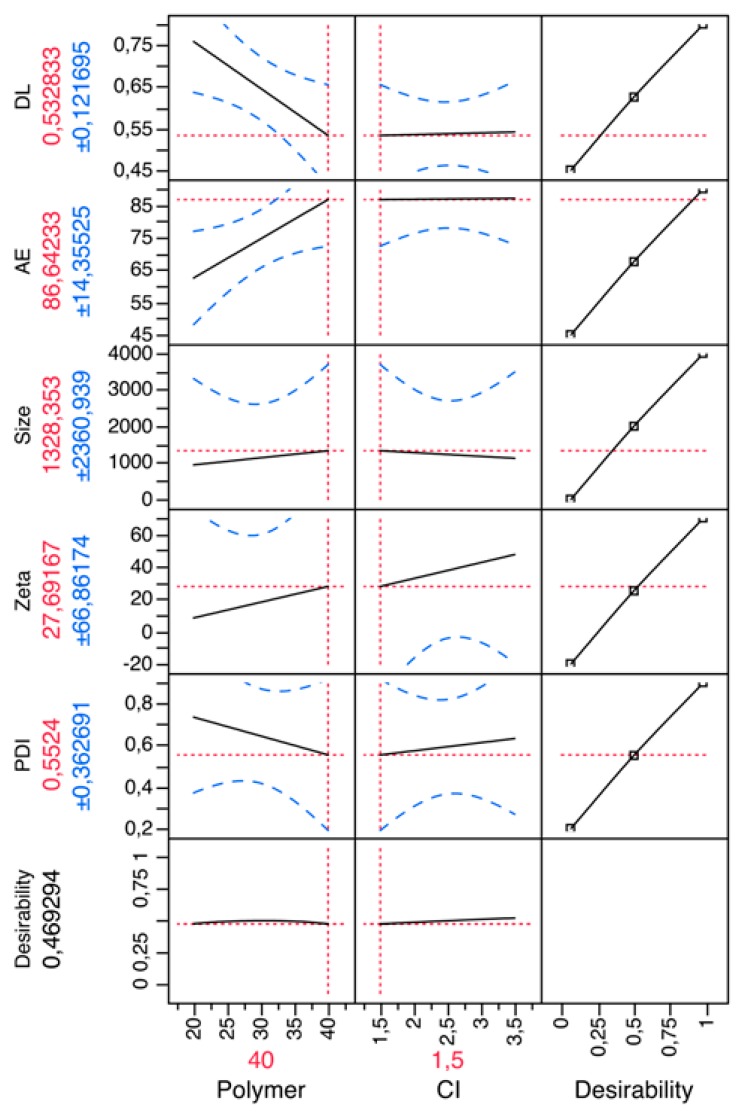
Prediction profiler for chitosan (CH) microparticles (MPs). X-axis: polymer (CH) (mg), counter-ion (TPP) (mg); Y-axis: polydispersity index (PDI), zeta potential (mV), size (nm), association efficiency (AE), drug loading (DL).

**Figure 2 pharmaceuticals-12-00032-f002:**
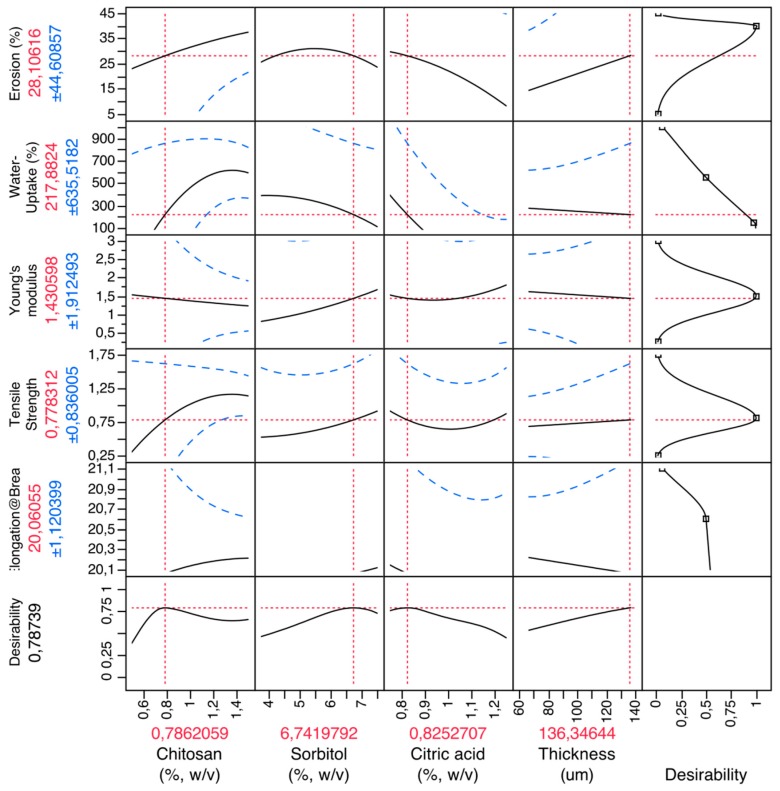
Prediction profiler for CH films. X-axis: excipients (chitosan, sorbitol, citric acid) and thickness; Y-axis: mechanical properties (elongation at break (%), tensile strength (MPa), Young’s modulus (MPa)), water uptake (%) and erosion (%).

**Figure 3 pharmaceuticals-12-00032-f003:**
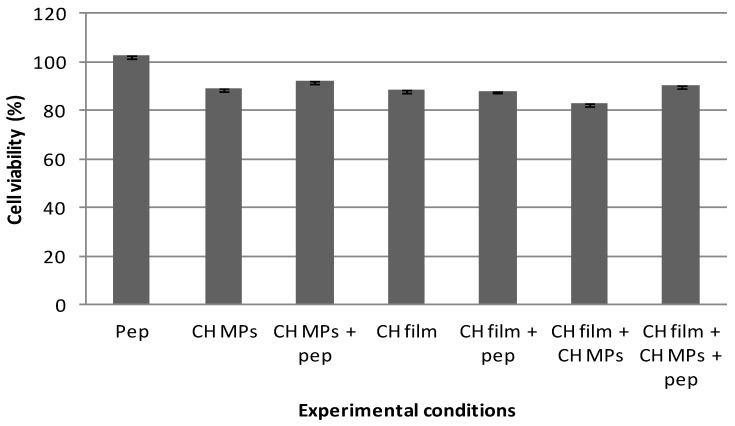
Cell viability under effect of peptide (5 μg/mL); CH MPs, CH film and CH MPs incorporated into CH film with and without peptide (5 μg/mL), measured by MTT assay and expressed as the mean ± SD (*n* = 5).

**Table 1 pharmaceuticals-12-00032-t001:** Characteristics of unloaded CH MPs and peptide-loaded CH MPs (CH MPs + peptide) (mean ± sd (*n* = 3)).

Caption	Size (µm)	Polydispersity Index	Zeta Potential (mV)	Association Efficiency (%)	Loading Capacity (%)
CH MPs	2.544 ± 0.97	0.66 ± 0.18	50.38 ± 7.18	-	-
CH MPs + Peptide	2.582 ± 0.87	0.45 ± 0.18	60.97 ± 9.20	76.16 ± 1.96	0.46 ± 0.01

**Table 2 pharmaceuticals-12-00032-t002:** Mechanical properties of CH films incorporated with peptide-loaded CH MPs.

Caption	Young’s Modulus (MPa)	Tensile Strength (MPa)	Elongation at Break (%)
CH MPs	2.12 ± 0.93	0.71 ± 0.09	20.06 ± 0.68
CH MPs + Peptide	2.29 ± 0.81	0.77 ± 0.09	20.27 ± 0.72

**Table 3 pharmaceuticals-12-00032-t003:** Swelling and erosion behavior of CH films incorporated with peptide-loaded CH MPs.

Erosion (%)	Swelling (%)	Disintegration Time (s)
20.03 ± 1.3	257 ± 56	30

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
