# Peer review of "Development and Characterization of Chitosan Microparticles-in-Films for Buccal Delivery of Bioactive Peptides"

_pharmaceuticals, 2019, doi:10.3390/ph12010032_

Round 1

Reviewer 1 Report

In general, is a very well written and supported paper. Minor corrections are suggested to the authors to improve the quality of the manuscript

Development and characterization of chitosan microparticles-in-films for buccal delivery of bioactive peptides

By Batista et al.

Abstract

Line 12: Remove the space or if something is missing after 2.5 __ m, please add it …” exhibited a mean diameter of 2.5  m, a…”

Line 18: Explain MTT. Include the main result of this assay.

In this section it is also important to include before the objective of the study, something related to the problem or opportunity to solve. Finally, the authors should close the abstract highlighting the impact of the study, not only in technical terms but also indicating the area or people that could be impacted with these results.

Introduction

I strongly recommend to include at the end of this section, the main areas that can be impact by the results of this research, not only in technical terms but also in terms of health or other terms. Also, if is possible to include a short paragraph with similar systems (polysaccharide and peptides included) to support with literature as well the application of this systems.

Material and methods

In the section 2.1 some reagents have the manufacturer and the country but they do not have city or state. Please add it.

In the section 2.2 it is important to highlight the temperature for the microparticles preparation.

Line 91-92: Please avoid the explanation of chemical interactions. It is a section where only methodological aspects should be written.

Lines 93-94: Include molar concentration instead of only milligrams amounts.

Line 103: It looks that something is missing, after the expression “independent variables”.

Line 119: “was used and UV detector wavelength was to 280 nm” it should be was set to 280 nm.

Section 2.5: Please add the molar amounts after the mass for each component and the molar ratios.

Lines 135-136: It is important to clarify the volume of the solution casted and the dimensions of the petri dishes.

Section 2.7: Add the molar amount after the mass. In this section is not clear the number of MPs added. Please clarify.

Results and discussion

Please, clarify legend in figure 1. In the legend you talk about TPP amount but in the figure, it is written: “surfactant/Cl”.

Line 258: The expression “The incorporation of the peptide the CH MPs did not have a significant effect on particle size” should be written better.

Please, improve the quality of table 1.

Line 290: Authors should discuss more about the loading efficiency as compare to other systems.

Line 302: Discuss more about the commercial viability and application of casting method to produce commercial films.

Line 316: What kind of properties the authors are referring?

Improve the quality of Tables 2 and 3.

Line 390: Please write “Figure” instead of “Fig.”

Could be used another test to assay biocompatibility?

Finally, I strongly recommend a general English edition, to improve the quality of the manuscript.

Author Response

Reviewer #1: 

This manuscript presents some innovative work on chitosan based materials for buccal delivery of bioactive peptides. Considering this innovative parameter it can be considered for publication in Pharmaceuticals. However corrections have to be done before publication.

Abstract
*       Line 12: Remove the space or if something is missing after 2.5 __ m, please add it …” exhibited a mean diameter of 2.5  m, a…”

A: Thank you for the remark. The correct denomination “2.5 µm” is now correctly used along the manuscript (please see the correction in line 14).

*       Line 18: Explain MTT. Include the main result of this assay.

A: Thank you for reviewer comment. The correct information “methylthiazolydiphenyl-tetrazolium bromide (MTT) assay with TR146 cells. The results indicating 75-99% cellular viability.” (please see the correction in lines 20-21).

*       Comment: In this section it is also important to include before the objective of the study, something related to the problem or opportunity to solve. Finally, the authors should close the abstract highlighting the impact of the study, not only in technical terms but also indicating the area or people that could be impacted with these results.

A: Thank you for the suggestion. Authors added the information to the manuscript and can be found in lines 8-10.

Introduction
*       Comment: I
strongly recommend to include at the end of this section, the main areas that can be impact by the results of this research, not only in technical terms but also in terms of health or other terms. Also, if is possible to include a short paragraph with similar systems (polysaccharide and peptides included) to support with literature as well the application of this systems.

A: Thank you for reviewer comment. Authors added the information to the manuscript and can be found in lines 78-84.

Materials and Methods

*       In the section 2.1 some reagents have the manufacturer and the country but they do not have city or state. Please add it.

A: Thank you for the suggestion. Authors added the information to the manuscript and can be found in lines 88, 99, 106 and 107.

*       In the section 2.2 it is important to highlight the temperature for the microparticles preparation.

A: Thank you for reviewer comment. The correct information is “at room temperature” was added to the manuscript and can be found in line 113.

*       Line 91-92: Please avoid the explanation of chemical interactions. It is a section where only methodological aspects should be written.

A: Thank you for reviewer comment. Authors removed the supplementary information (please see the correction in line 110).

*       Line 93-94: Include molar concentration instead of only milligrams amounts.

A: Thank you for reviewer comment. Molar concentration information was added to the manuscript and can be found in line 110.

*       Line 103: It looks that something is missing, after the expression “independent variables”.

Include molar concentration instead of only milligrams amounts.

A: Thank you for the remark. However, since the sentence was not clear, authors tried to improving it aiming to clarify the statement and for the text to become more reader-friendly (please check lines 121 and 122).

*       Line 119: “was used and UV detector wavelength was to 280 nm” it should be was set to 280 nm.

A: Thank you for reviewer comment. The correct information “UV detector wavelength was set to 280 nm” was added to the manuscript (please check lines 137).

*       Section 2.5: Please add the molar amounts after the mass for each component and the molar ratios.

A: Thank you for reviewer comment. Authors added the information to the manuscript and can be found in lines 156-158.

*       Line 135-136: It is important to clarify the volume of the solution casted and the dimensions of the petri dishes.

A: Thank you for the remark. The correct information “10mL” and “Petri dishes (90x15 mm) was added to the manuscript (please check lines 160).

*       Section 2.7: Add the molar amount after the mass. In this section is not clear the number of MPs added. Please clarify.

A: Thank you for reviewer comment. Authors added the information to the manuscript and can be found in lines 209 and 210.

Results and Discussion

*       Please, clarify legend in figure 1. In the legend you talk about TPP amount but in the figure, it is written: “surfactant/Cl.

A: Thank you for noticing this mistake. The correct denomination “CI (counter-ion)” is now correctly (please see the correction in line 274.

*       Line 258: The expression “The incorporation of the peptide the CH MPs did not have a significant effect on particle size” should be written better.

A: Thank you for reviewer comment. However, since the sentence was not clear, authors tried to improving it aiming to clarify the statement and for the text to become more reader-friendly (please check lines 294-296).

*       Please, improve the quality of table 1.

A: Thank you for reviewer comment. Authors tried to improving the quality of table 1 (please check lines 299).

*       Line 290: Authors should discuss more about the loading efficiency as compare to other systems.

A: The authors acknowledge the suggestion and added information. The alteration can be found at lines 326-331).

*       Line 302: Discuss more about the commercial viability and application of casting method to produce commercial films.

A: Thank you for the interesting question. Authors added the information can be found at lines 352-354).

*       Line 316: What kind of properties the authors are referring?

A: Thank you for the comment. Authors added the information (please check lines 371-372).

*       Improve the quality of Table 2 and 3.

A: Thank you for reviewer suggestion. Authors tried to improving the quality of table 2 and 3 (please check lines 392 and 438).

*       Line 390: Please write “Figure” instead of “Fig.”

A: Thank you for noticing this mistake. Authors corrected the designation of “Fig.” incorrectly added to “Figure” (please see the correction in lines 453).

*       Could be used another test to assay biocompatibility?

A: Thank you for the interesting question. Yes, we confirm de MTT results with the LDH assay.

Reviewer #2: 

Dear reviewer, the authors acknowledge the suggestions made to improve the quality of the text.

*       Line 257: The text is a little confusing.  The authors appear to suggest that 'the encapsulation method induced a slight increase in particle size', then go on to state that 'incorporation of the peptide the CH MPs did not have a significant effect'.  I agree: the standard deviations indicated no statistical difference between the means.  Indeed, none of the parameters listed in Table 1 were significantly different, at the level of 2 standard deviations.  I suggest the authors should modify their text to reflect that

A: The authors acknowledge the suggestion to make the sentence clearer and therefore proceeded to the alteration than can be found at lines 294-296.

*       Line 373: In view of the uncertainty - especially for the % swelling (quoted as 56.57), I suggest it not necessary to quote the results to 2 decimal places.

A: The authors acknowledge the suggestion can be found in line 438.

Typographical errors:

*       Line 12: A character seems to be missing in the manuscript: '...mean diameter of 2.5 µm....'

A: Thank you for noticing this mistake. The correct denomination is “2.5 µm” (please see the correction in line 14).

*       Line 54: A word or phrase seems to be missing '...because have shown to posess...' (...because previous studies have shown the material to posess...?)

A: Thank you for the remark. Authors clarified the sentence (please see the correction in line 59).

*       Line 188: Something seems to be missing from the phrase '...for examination the biological functionality...' (of the biological functionality?)

A: Thank you for reviewer comment. Authors tried to improve the sentence aiming to clarify the statement and for the text to become more reader-friendly (please check line 217).

*       Line 262: A character seems to be missing from the 'Size' column heading of table 1.

A: Thank you for noticing this mistake. The correct denomination is “µm” (please see the correction in line 300).

*       Line 309, Line 336 (table 2), Line 347: Wrong spelling of 'brake' - I think it should be 'break'.

A: Thank you for noticing this mistake. The correct denomination “break” is now correctly used along the manuscript (please see the correction in lines 363, 392, 403 and 419).

*       Actually, what is 'Elongation break'?  Do you mean elongation to break?  If so, the units should be strain (a number or per-centage) - not MPa.

A: Thank you for noticing this mistake. The correct unit is a percentage and is not MPa (please check in line 392).

*       Line 331-333: Something seems to be missing from the sentence 'So it is needed to evaluate mechanical properties because determine films integrity and performance during utilization and storage'.

A: Thank you for the comment. However, since the sentence was not clear, authors tried to improve it aiming to clarify the statement and for the text to become more reader-friendly (please check lines 387-389).

*       Line 350-351: '...were low than compared with other oral films...' should be: '...were lower than other oral films...' ot '...were low, when compared with other oral films...'

A: Thank you for suggestion. Authors changed the phrase in order to be clearer (please see de correction in lines 406-416).

*       Line 359-360: The phrase '...they are immediately hydrated by saliva following disintegration and/or dissolution...' does not make sense.  I think the authors may mean '...they are immediately hydrated by saliva, followed by disintegration and/or dissolution...

A: The authors acknowledge the suggestion to make the sentence clearer and therefore proceeded to the alteration than can be found at line 425.

*       Line 360-364: I suggest the next sentence '...This CH film with peptide-loaded CH MPs when contacted...' is also confusing, did not make very good reading and should be re-worded.

A: Thank you for the comment. However, since the sentence was not clear, authors tried to improve it aiming to clarify the statement and for the text to become more reader-friendly. This alteration can be found at lines 425-427).

*       Line 365: '...which facilitate the rapid...' should be '...which facilitates the rapid...'

A: The authors thank the suggestion that can be found at line 428.

*       Line 368: There is a grammatical error in the phrase '...data obtained in this study confirm the reported in the literature...'  Do the authors mean '...data obtained in this study confirm other reports in the literature...''

A: Thank you for noticing this mistake. Authors corrected the sentence that can be found at line 433.

*       Line 390: '...assay of TR146 cells which after being exposed....' should be '...assay of TR146 cells after exposure....'

A: The authors thank the suggestion that can be found at line 453.

*       Line 404: I suggest '...gelation method and it showed desirable pharmaceutical...' should be '...gelation method and showed desirable pharmaceutical...'

A: The authors acknowledge the suggestion to make the sentence clearer and proceeded to the modification (please see the correction in line 485.

*       Line 415: 'founds' should be 'funds'.

A: Thank you for noticing this mistake. Authors corrected the sentence that can be found at line 496.

Reviewer 2 Report

This manuscript describes the formation and testing of chitosan-based films for buccal delivery of bio-active peptides.  In my opinion, although it does not explore the underlying science in great detail, this manuscript does present some useful information, which merits publication.

I have only a couple of comments on the technical content:

Line 257: The text is a little confusing.  The authors appear to suggest that 'the encapsulation method induced a slight increase in particle size', then go on to state that 'incorporation of the peptide the CH MPs did not have a significant effect'.  I agree: the standard deviations indicated no statistical difference between the means.  Indeed, none of the parameters listed in Table 1 were significantly different, at the level of 2 standard deviations.  I suggest the authors should modify their text to reflect that.

L373: In view of the uncertainty - especially for the % swelling (quoted as 56.57), I suggest it not necessary to quote the results to 2 decimal places.

The manuscript also suffers from a considerable number of typographical errors::

L12: A character seems to be missing in the manuscript: '...mean diameter of 2.5 m....'

L54: A word or phrase seems to be missing '...because have shown to posess...' (...because previous studies have shown the material to posess...?)

L188: Something seems to be missing from the phrase '...for examination the biological functionality...' (of the biological functionality?)

L262: A character seems to be missing from the 'Size' column heading of table 1.

L309, L336 (table 2) L347: Wrong spelling of 'brake' - I think it should be 'break'.

Actually, what is 'Elongation break'?  Do you mean elongation to break?  If so, the units should be strain (a number or per-centage) - not MPa.

L331-333: Something seems to be missing from the sentence 'So it is needed to evaluate mechanical properties because determine films integrity and performance during utilization and storage'.

L350-351: '...were low than compared with other oral films...' should be: '...were lower than other oral films...' ot '...were low, when compared with other oral films...'

L359-360: The phrase '...they are immediately hydrated by saliva following disintegration and/or dissolution...' does not make sense.  I think the authors may mean '...they are immediately hydrated by saliva, followed by disintegration and/or dissolution...'

L360-364: I suggest the next sentence '...This CH film with peptide-loaded CH MPs when contacted...' is also confusing, did not make very good reading and should be re-worded.

L365: '...which facilitate the rapid...' should be '...which facilitates the rapid...'

L368: There is a grammatical error in the phrase '...data obtained in this study confirm the reported in the literature...'  Do the authors mean '...data obtained in this study confirm other reports in the literature...'

L390: '...assay of TR146 cells which after being exposed....' should be '...assay of TR146 cells after exposure....'

L404: I suggest '...gelation method and it showed desirable pharmaceutical...' should be '...gelation method and showed desirable pharmaceutical...'

L415: 'founds' should be 'funds'.

Author Response

Dear Editor of Pharmaceuticals Journal,

We deeply acknowledge the interest demonstrated in our work and the availability to reconsider a revised version of this manuscript. 

We undertook several modifications in the manuscript according the comments of Reviewers, in order to improve its overall quality and fulfil the requirements of Pharmaceuticals Journal.

Below we provide point-by-point answers to the comments on the manuscript. All changes performed to the manuscript are highlighted in the revised version.

*

Reviewer #2: 

Dear reviewer, the authors acknowledge the suggestions made to improve the quality of the text.

*       Line 257: The text is a little confusing.  The authors appear to suggest that 'the encapsulation method induced a slight increase in particle size', then go on to state that 'incorporation of the peptide the CH MPs did not have a significant effect'.  I agree: the standard deviations indicated no statistical difference between the means.  Indeed, none of the parameters listed in Table 1 were significantly different, at the level of 2 standard deviations.  I suggest the authors should modify their text to reflect that

A: The authors acknowledge the suggestion to make the sentence clearer and therefore proceeded to the alteration than can be found at lines 294-296.

*       Line 373: In view of the uncertainty - especially for the % swelling (quoted as 56.57), I suggest it not necessary to quote the results to 2 decimal places.

A: The authors acknowledge the suggestion can be found in line 438.

Typographical errors:

*       Line 12: A character seems to be missing in the manuscript: '...mean diameter of 2.5 µm....'

A: Thank you for noticing this mistake. The correct denomination is “2.5 µm” (please see the correction in line 14).

*       Line 54: A word or phrase seems to be missing '...because have shown to posess...' (...because previous studies have shown the material to posess...?)

A: Thank you for the remark. Authors clarified the sentence (please see the correction in line 59).

*       Line 188: Something seems to be missing from the phrase '...for examination the biological functionality...' (of the biological functionality?)

A: Thank you for reviewer comment. Authors tried to improve the sentence aiming to clarify the statement and for the text to become more reader-friendly (please check line 217).

*       Line 262: A character seems to be missing from the 'Size' column heading of table 1.

A: Thank you for noticing this mistake. The correct denomination is “µm” (please see the correction in line 300).

*       Line 309, Line 336 (table 2), Line 347: Wrong spelling of 'brake' - I think it should be 'break'.

A: Thank you for noticing this mistake. The correct denomination “break” is now correctly used along the manuscript (please see the correction in lines 363, 392, 403 and 419).

*       Actually, what is 'Elongation break'?  Do you mean elongation to break?  If so, the units should be strain (a number or per-centage) - not MPa.

A: Thank you for noticing this mistake. The correct unit is a percentage and is not MPa (please check in line 392).

*       Line 331-333: Something seems to be missing from the sentence 'So it is needed to evaluate mechanical properties because determine films integrity and performance during utilization and storage'.

A: Thank you for the comment. However, since the sentence was not clear, authors tried to improve it aiming to clarify the statement and for the text to become more reader-friendly (please check lines 387-389).

*       Line 350-351: '...were low than compared with other oral films...' should be: '...were lower than other oral films...' ot '...were low, when compared with other oral films...'

A: Thank you for suggestion. Authors changed the phrase in order to be clearer (please see de correction in lines 406-416).

*       Line 359-360: The phrase '...they are immediately hydrated by saliva following disintegration and/or dissolution...' does not make sense.  I think the authors may mean '...they are immediately hydrated by saliva, followed by disintegration and/or dissolution...

A: The authors acknowledge the suggestion to make the sentence clearer and therefore proceeded to the alteration than can be found at line 425.

*       Line 360-364: I suggest the next sentence '...This CH film with peptide-loaded CH MPs when contacted...' is also confusing, did not make very good reading and should be re-worded.

A: Thank you for the comment. However, since the sentence was not clear, authors tried to improve it aiming to clarify the statement and for the text to become more reader-friendly. This alteration can be found at lines 425-427).

*       Line 365: '...which facilitate the rapid...' should be '...which facilitates the rapid...'

A: The authors thank the suggestion that can be found at line 428.

*       Line 368: There is a grammatical error in the phrase '...data obtained in this study confirm the reported in the literature...'  Do the authors mean '...data obtained in this study confirm other reports in the literature...''

A: Thank you for noticing this mistake. Authors corrected the sentence that can be found at line 433.

*       Line 390: '...assay of TR146 cells which after being exposed....' should be '...assay of TR146 cells after exposure....'

A: The authors thank the suggestion that can be found at line 453.

*       Line 404: I suggest '...gelation method and it showed desirable pharmaceutical...' should be '...gelation method and showed desirable pharmaceutical...'

A: The authors acknowledge the suggestion to make the sentence clearer and proceeded to the modification (please see the correction in line 485.

*       Line 415: 'founds' should be 'funds'.

A: Thank you for noticing this mistake. Authors corrected the sentence that can be found at line 496.

*

The authors would like to acknowledge the reviewer for the availability demonstrated to revise this manuscript. We also acknowledge the careful reading and the enriching comments performed that helped us improving the quality of the work.

We trust that we have carefully and appropriately addressed all the reviewer’s questions and concerns and thank you for your interest in this manuscript.

Sincerely yours,

Manuela Pintado
